# Children and young people's beliefs about mental health and illness in Indonesia: A qualitative study informed by the Common Sense Model of Self-Regulation

Helen Brooks[1]*, Kirsten Windfuhr[1], Irmansyah[2], Benny Prawira[3,4], Dyah Afina Desyadi Putriningtyas[4], Karina Lovell[1,5], Susi Rutmalem Bangun[6☯], Armaji Kamaludi Syarif[2☯], Christa Gumanti Manik[2☯], Ira Savitri Tanjun[7☯], Soraya Salim[8], Laoise Renwick[1], Rebecca Pedley[1], Penny Bee[1]

1 Division of Nursing, Midwifery and Social Work, School of Health Sciences, Manchester Academic Health Science Centre, University of Manchester, Manchester, United Kingdom, 2 National Institute of Health Research and Development, Ministry of Health, Jakarta, Republic of Indonesia, 3 Atma Jaya Catholic University of Indonesia, Jakarta, Republic of Indonesia, 4 Into the Light, Jakarta, Republic of Indonesia, 5 Greater Manchester Mental Health NHS Foundation Trust, Manchester, United Kingdom, 6 Soerojo Mental Health Hospital, Magelang, Republic of Indonesia, 7 Marzoeki Mahdi Hospital, Bogor, Republic of Indonesia, 8 Pulih@the Peak–Women, Youth and Family Empowerment Centre, Jakarta, Republic of Indonesia

☯ These authors contributed equally to this work.
* helen.brooks@manchester.ac.uk

**Data Availability Statement:** Due to the potentially identifying nature of the transcripts, complete transcripts cannot be made publicly available.

## Abstract

### Background

Mental illness is a leading cause of disease burden amongst children and young people (CYP). This is exacerbated in low- and middle-income (LMIC) countries which often have embryonic care structures. Understanding and targeting illness beliefs is a potentially efficacious way of optimising the development of health prevention interventions. These beliefs remain relatively underexplored in CYP in LMIC contexts. Aim: To develop an in-depth understanding of CYPs beliefs about mental health and illness in Indonesia.

### Methods and findings

Semi-structured interviews (n = 43) combined with photo elicitation methodology were undertaken with CYP aged 11–15 from Java, Indonesia. Our sample comprised those living with (n = 19) and without (n = 24) high prevalence mental health conditions, specifically anxiety or depression. Data were analysed using framework analysis, informed by the Common Sense Model of Self-Regulation of Health and Illness. Positive mental health and illness were dichotomised in accounts with mental health typically characterised as an absence of mental disturbance. This contributed to attributions of abnormality and the marginalisation of those with mental illness. Mental illness was conceptualised as a single entity, commonly arising from individual failings. This prompted feelings of self-stigma in those with lived experience of mental illness. Analysis identified marked differences in the perceived time dimensions of positive mental health and illness with mental illness conceived as less transient

Public Access is restricted by the University of Manchester Research Ethics Committee (Ref: 2018–4949-7908) and The Ministry of Health Indonesia (Ref: LB:02.01/2/KE.201/2019). All relevant data are available upon request (subject to compliance with current ethical permissions) to the information governance representatives of the University of Manchester at: information.governance@manchester.ac.uk.

**Funding:** This authors who received funding were HB, II, BP, KL, IST, LR, PB. This project is funded by the Medical Research Council/Department for International Development/National Institute for Health Research/Economic and Social Research Council's programme of research to improve adolescent health in low and middle income countries: https://mrc.ukri.org/funding/science-areas/global-health-and-international-partnerships/funding-partnerships/adolescent-health/ Award number: MR/R022151/1. The funders and sponsor have no role in study design, data collection and analysis, decision to publish, or preparation of manuscripts.

**Competing interests:** The authors have declared that no competing interests exist.

**Abbreviations:** CSM, Common sense model of self-regulation; CYP, Children and young people; LMIC, low and middle income countries; UK–, United Kingdom; PPIE, patient and public involvement and engagement; WHO, World Health Organisations.

than episodes of positive mental health. Illness beliefs appeared relatively consistent across the two groups of CYP although some nuanced differences were identified. CYP with anxiety and depression were less likely to believe that mental illness could be diagnosed visually, more likely to uphold multiple causal factors and endorse the potential efficacy of professional input.

## Conclusions

Public health interventions to increase understanding may be necessary to develop healthcare systems to reduce treatment barriers, optimise return on investment and enhance population health effect.

## Introduction

Mental illnesses account for a significant proportion of the global burden of disease amongst children and young people (CYP) [1]. Globally, between 10–20% of CYP experience mental health problems although the majority go undiagnosed and untreated [2, 3]. Given that the majority of adult disorders manifest before the age of 18, this developmental period is critical in identifying and treating mental illness to reduce current and prevent future disease burden [4]. The impact of experiencing childhood mental health problems is long-lasting. Depression during adolescence is associated with more severe forms of mental illness during adulthood, increased risk of suicide, poorer physical and psychosocial outcomes and reduced quality of life [5, 6].

CYP in low- and middle-income countries (LMICs) are at increased risk of developing mental health problems such as anxiety and depression [7]. They are also more likely to experience adverse outcomes due to a range of socioeconomic factors such as higher levels of poverty, greater exposure to conflict and natural disasters [6]. Further, there is evidence to suggest approximately 90% of people living in LMICs remain unable to access services due to organisational level barriers including a lack of adequate funding, insufficient service infrastructure and a lack of appropriately trained mental health workforce [8–10]. Despite recent calls from the World Health Organisation (WHO) for the development and upscaling of services for this group, significant treatment gaps persist [11].

More recently criticism has been levelled at the narrow definition of treatment which underpins the treatment gap paradigm. Most notably this criticism has focused on the inherent dominance of biomedical connotations and a perceived inability to consider the wide variety of illness experiences and treatment options which may be considered by people experiencing distress in LMICs [12]. Healthcare access is likely to be a result of a complicated interplay of top-down service provision and ground level beliefs and behaviours including individual recognition and help seeking behaviours. Such concerns have directed attention towards alternative foci, such as resource optimization, appropriate and holistic care navigation and self-management approaches, which emphasise the need to understand individual concepts of health and healthcare acceptability. Evidence suggests that CYP beliefs about and attitudes towards mental health and illness are more important predictors of help-seeking behaviours than structural barriers [13].

Indonesia, an archipelago in South East Asia, is the fourth most populated country in the world with an estimated population of 268 million people; including 68 million adolescents [14]. Evidence suggests that almost 50% of high school students experience depressive

symptoms and that approximately 10% of 15–24-year-olds have a diagnosis of non-specific emotional disorder [15]. Research in adult and child populations in Indonesia has highlighted the salience of mystical and supernatural forces as perceived determinants of mental illness and demonstrated the negative impact of stigma on conceptualisations of health and illness [16, 17].

Despite the recent re-classification of Indonesia as a middle income country according to World Bank Classifications, and the recognition of mental health as a national priority [18], community mental health services remain mostly embryonic with variable and limited coverage [19]. The recent national prioritisation of mental health represents a potential and timely opportunity to develop interventions which support health system development but also meet the needs of CYP in Indonesia.

The role of illness perceptions and their relationship to health-related behaviours and outcomes have been studied extensively in people with physical health conditions. The widely used Common Sense Model of Self-Regulation of Health and Illness (CSM) provides a multi-level conceptual framework which explicates the processes by which an individual develops an understanding of threats to their health. This understanding is achieved by drawing on social and experiential information and is thought to influence how people view symptoms, enact help seeking behaviours and engage with self-management and treatment options [20]. Previous research has identified five components which are thought to encapsulate people's illness perceptions: 1) beliefs about the identity of illness, 2) causal attributions, 3) timeline dimensions, 4) curability and 5) consequences and one non cognitive domain: emotional perceptions [21]. Illness perceptions are thought to be dynamic and responsive to increased illness experience and the assimilation of related information and experiences over time. As such cognitive representations of illness are iterative and develop in a form of cyclical loop as understandings of conditions develop [22]. Dysfunctional illness perceptions have been associated with longer recovery periods, higher use of health services and increased levels of disability for people with both physical and mental health conditions [23, 24].

Whilst such approaches are comparatively underexplored in the field of mental health, there have been recent attempts to explore the application of the CSM for people with a diagnosis of mental illness. To date, findings have been equivocal with research demonstrating both utility of the CSM for people with mental health problems [25, 26] as well as significant differences in illness perceptions between those with mental and physical health problems [27, 28]. Extant research has also tended to focus on how people interpret experiences (both internal and external) and how these are perceived to relate to initiating and sustaining mental health symptoms. This has been to the detriment of in-depth understandings of the everyday experience of living with a mental health problem [26] and broader conceptualisations of well-being [29]. There has been limited application of the CSM to CYP populations, particularly those living in LMIC contexts. However, available evidence in the form of quantitative studies with CYP aged between 12–17 [30] and mixed methods studies with young adults aged 18–25 [31] from high income countries demonstrate the salience of this model to CYP populations. This highlights the need for further exploratory work to examine the cultural and developmental transferability of the CSM.

Understanding the perceptions and attitudes that CYP hold towards mental health is crucial to optimising the development of acceptable and targeted interventions. The present study adopts a broad interpretation of health perceptions and includes both perceptions of positive mental health and mental illness [17]. Further, this study takes account of criticisms of traditional approaches which often fail to take social cultural factors into consideration [29]. This study builds on existing evidence by combining semi-structured one-to-one interviews and photo elicitation methods with an existing conceptual framework (CSM) to develop an in-

depth understanding of mental health and illness perceptions amongst 11–15 year olds in Java, Indonesia whilst also examining the impact of lived experience on identified perceptions. This age range was decided upon following early consultation with Indonesian collaborators because it represents the junior high school population in Indonesia.

## Materials and methods

### Participants and setting

The study was conducted in three sites in Java, Indonesia: Jakarta, Bogor and Magelang. Sites were selected due to the fact they differed on key variables including culture, urbanisation and health service development. These three sites also had established child and adolescent mental health services which were important to support study recruitment. The data presented in this manuscript forms part of a larger programme of work which aims to co-develop and evaluate an anxiety and depression focussed mental health literacy intervention for children and young people aged 11–15 [32].

Participants were purposively sampled on age, gender, geographical location and whether or not they had experience of common mental health problems, namely anxiety and depression. Recruitment continued until the research team reached consensus that data saturation had been reached and there were no new themes arising from the data [33]. In order to be eligible to take part in the study, participants had to be aged between 11 and 15 years of age and living in one of the three study sites (Jakarta, Bogor and Magelang). See Table 1 for further information on study participants.

|CYP with a diagnosis of anxiety or depression were invited directly by child and adolescent mental health services. CYP with no experience of mental health problems were recruited through direct invitation by schools. Posters were also exhibited in relevant organisations to supplement the recruitment process and participants could contact the study team directly if they were interested in taking part in the study. However, all CYP were recruited through direct invitation.

**Table 1. Demographic information.**

|  | % (n) |
|---|---|
| **Gender** | |
| Male | 22 (51%) |
| Female | 21 (49%) |
| **Age** | |
| *Mean* | 12.8 years |
| *Range* | 11–15 |
| *11* | 26% (n = 11) |
| *12* | 16% (n = 7) |
| *13* | 23% (n = 10) |
| *14* | 21% (n = 9) |
| *15* | 14% (n = 6) |
| **Area** | |
| Jakarta | 30% (n = 13) |
| Bogor | 33% (n = 14) |
| Magelang | 37% (n = 16) |
| **Experience of mental illness** | |
| Yes | 44% (n = 19) |
| No | 56% (n = 24) |

Ethical approval for the study was granted by University of Manchester Research Ethics Committee (Ref: 2018–4949–7908) and The Ministry of Health Indonesia (Ref: LB:02.01/2/KE.201/2019).

## Data collection

Semi-structured interviews were combined with photo elicitation methodology in order to facilitate discussion around the potentially sensitive subject of mental health and illness in Indonesia. Such methods have been shown to promote researcher-participant rapport during interviews, leading to new perspectives and increased participant empowerment during the research process in both adult and CYP populations [34, 35].

Participants expressing an interest in taking part in the study contacted the research team directly and were invited along with their parents to attend an initial meeting with the study team. CYP participants were provided with an age-appropriate information sheet, along with a crib sheet detailing the photo elicitation methods. Parents were provided with separate information sheets. Both CYP and their parents were given the opportunity to ask any questions about the study prior to making a decision about participation. Interviews only proceeded if CYP provided informed verbal assent to take part and their parent/carer provided informed written consent for their child's participation.

The crib sheet invited participants to bring photographs along to the interview which represented 'what mental health means to you' (see S1 Fig for more detail on how the task was described to potential participants). Participants were informed that taking photographs was an optional part of the study; participants remained free to take part in an interview if they chose not to take photographs. It was made clear to participants that they would be asked to share and discuss any photographs taken during their interview. Participants were advised to use their own smartphone and that if they did not have one available for them to use, one would be provided for the purposes of taking part in the study. No participants requested access to a study specific mobile phone.

Data was collected in quiet and private rooms in either schools, outpatient clinics or community organisations by Indonesian researchers who were trained and supervised by senior Indonesian and UK academics with significant experience in qualitative data collection and analysis. To encourage CYP to be able to speak openly, parents were encouraged to not attend the interview. Interviews started with questions about the photographs they had taken and participants were asked to explain what they had meant to convey with their pictures and how this related to their understanding of mental health and illness. The remainder of the interview followed an interview schedule designed to elicit beliefs about positive mental health and wellbeing and mental illness. Questions explored CYP's perceptions of mental health in general, with particular focus on anxiety and depression. Example questions included:

- What do you think 'mental health' means (Prompt good mental health vs. mental health problems)?

- What do you think it means when people have a 'mental health problem' or 'mental illness'?

- What do you think the signs of mental health problems are?

- What do you think the causes of mental health problems are?

- Is there anything that young people can do for themselves to look after their mental health?

- What sort of things can help people with mental health problems?

Interviews ranged between 20 and 40 minutes in length and were undertaken in Bahasa Indonesian by health services researchers at the Ministry of Health in Indonesia or at local sites who received training and ongoing supervision in qualitative research. Interviews were recorded using encrypted digital recorders before being transcribed verbatim in Bahasa and anonymised. Transcripts were translated into English by a member of the research team. 5% of translated transcripts were checked by a third party not involved in data collection to identify any discrepancies in meaning compared to the original transcript in accordance with the guidelines for undertaking international qualitative research.

## Analysis

Photographs provided by CYP were not treated as units of data but as tools to support CYP participation in interviews [36] and promote rapport between the interviewer and interviewee [37]. Such approaches have been found to be particularly useful when discussing sensitive topics with younger populations [38].

Transcripts were analysed following an inductive framework approach [39]. Framework analysis incorporates five phases which involves researchers firstly familiarising themselves with the data by studying interview transcripts before undertaking initial coding. An initial framework of health and illness beliefs was developed from the codes from the first four interviews which was subsequently applied to the remaining transcripts. Regular analysis meetings were held between analysts and the wider study team to identify any required modifications to the initial framework that arose from the constant comparison of new data and more developed understanding of included data. Identified beliefs were then mapped to the five components of the CSM: illness identity, causal attributions, timeline dimensions, curability and consequences. The indexing and charting process was supported by Excel with initial categories assigned column headings and rows/sheets populated with demographic information. The analysis process culminated in a final framework which was considered representative of the dataset as a whole.

## Rigour

Reflexivity is an integral feature of qualitative analysis and the relationship between the researchers, the emerging analysis and the participants were examined in-depth during the analytical process [40]. The research team reported no predefined expectations relating to data analysis that they brought to the study above an interest in furthering current understandings of mental health from the perspective of children and young people in Indonesia. We also incorporated various strategies to enhance the trustworthiness and transparency of data analysis including transcripts being independently coded by KW who had no prior knowledge of or involvement in the study. HB and KW met after the independent coding of four transcripts to ensure consistency in coding. Interpretations and emerging analysis were discussed with the wider research team including both UK and Indonesian researchers to further enhance the credibility of analysis and ensure interpretations were not overly affected by individual members of the research team. Data and preliminary analysis were presented to a Patient and Public Involvement and Engagement (PPIE) advisory group made up of CYP and adults with lived experience of anxiety and depression and other relevant stakeholders (parents, teachers, health professionals and CYPs) at a four-day workshop in January 2020 to further verify interpretations and ensure the analysis was grounded in the lived experience of anxiety and depression in Indonesia.

## Results

43 children and young people consented to take part in the study (19 with experience of high prevalence mental health conditions namely anxiety and depression and 24 without such

experience); 31 out of the 43 participants provided photos upon presentation at the interview. Examples of photos included landscapes, the participant themselves, other people (friends, family and community members) photos of printed images and images taken directly from the internet.

Participants with and without experience of mental illness found it difficult to articulate what mental health and illness represented to them initially, although this was particularly apparent for CYP without experience of mental illness. The latter reported limited experience of mental illness either personally or within their family and wider communities, which meant they had limited knowledge, language or awareness of included concepts or readily available explanatory frameworks on which to draw to articulate their thoughts.

The results are presented under the six CSM components: identify beliefs, emotional representation of illness, causal attributions, timeline dimensions, beliefs about curability and beliefs about consequences. Within each section the similarities and differences identified between those with and without experience of mental health problems are presented which were the focus of our comparative analysis. A summary diagram is also presented in S1 Fig. No differences in study data were discernible in relation to gender, age or geographical location.

## Identity beliefs

Positive mental health and illness were dichotomised within the accounts of both groups of participants. Positive mental health was often characterised in both groups of participants' articulations by the absence of mental illness or an absence of the perceived key features of mental illness.

*Crazy people aren't healthy and aren't sane*. **ID34, Magelang, no experience of mental illness.**

*I think mental health is being healthy and not depressed and not having any other mental illness.* **ID10, Jakarta, experience of mental illness.**

Both groups coalesced in their attributions of mental illness as a single concept rather than a heterogeneous set of different conditions. For CYP without a diagnosis of anxiety or depression, there was a strong feeling that there were generic features of mental illness that are somaticized and externally observable to others through abnormal emotional expression or physical and behavioural manifestations. As such, there was a belief within these participants' accounts that mental illnesses could be readily diagnosed by professionals using visual methods.

*They can 'correct' our mental health problems and we can find out what's wrong with it. Usually psychiatrists or psychologists can know about it at first sight, maybe from [the patient's] eyes or their souls.* **ID28, Magelang, no experience of mental illness.**

*Someone [with mental health problems] is disturbed mentally, always sad by themselves or just always being alone by themselves, like we don't know what they're doing.* **ID14, Bogor, no experience of mental illness.**

For CYP with mental health experience, most did not feel that visual diagnosis of mental illness was possible. Instead, they felt they would need to be told by the person themselves or by a close family member if someone had a mental illness. A minority of CYP with a diagnosis of anxiety and depression described a heightened awareness and empathy towards people with mental illness because of their own experiences but this did not include or extend to a capacity to physically or visually identify mental illness.

*They have the same problems that I do, I can feel it, I can recognize it.* **ID10, Jakarta, experience of mental illness.**

## Emotional representations of illness

Given the perceived dichotomy between positive mental health and illness, those with mental health problems were often marginalised, treated as insignificant or peripheral, in interviews. This was evident in interviews with those with and without experience of mental illness. People with a mental illness were often referred to in interviews as 'abnormal' and on rarer occasions as dangerous people to be avoided. For CYP with experience of mental illness, this marginalisation appeared to lead to increased levels of self-stigma due to their own experiences.

*People with mental illness look different. Normal people are more focused, while mentally ill people sometimes wouldn't be able to focus, they'd laugh a lot and try to get others' attention.* **ID28, Magelang, no experience of mental illness.**

*When I see my friends, I see that most of them are normal, and they can do things well. But when I talk to other people, both of my hands tremble. I feel embarrassed.* **ID10, Jakarta, experience of mental illness.**

When talking about their own condition, accounts of mental health problems as shameful, embarrassing and burdensome to others were salient in CYP accounts. The potential for disclosure to lead to ridicule from others was often highlighted.

*Q: If you have a mental health problem like anxiety, depression, will you seek help?*

*R: Yes, I'd go to my mom. It's hard to seek help because I feel shameful.* **ID3, Magelang, no experience of mental illness.**

*Q: How easy or difficult is it for you to tell someone else that you have a problem? Please elaborate.*

*R: It's hard because I'm afraid that they'll mock me. Sometimes I get embarrassed to talk to someone else.* **ID10, Jakarta, experience of mental illness.**

## Causal attributions

Participants in both groups identified a range of similar stressors and external events that could trigger the onset of mental illness. However, there was a shared perception that despite these identified stressors, the ultimate cause of mental illness was as a result of individual weakness. Internal strength was perceived as a fixed characteristic, something that people either did or did not possess and something that was considered lost or absent when mental illness was present.

*Mental health is the health of a person, namely their feelings. "Mental capability" is something that a person has to be able to face things in their life, like a problem or something. To be brave, to be able to face something that's difficult for them. For example, whether or not you're able to go in front of the classroom to answer a question is a barometer of how weak or strong your mental [health] is. Some people have a courageous mental capability, some don't.* **ID28 Magelang, no experience of mental illness.**

*Q: Why do you think people are depressed or anxious?*

*R: Errr, because they have a problem in their life.*

*Q: How does it happen?*

*R: They keep thinking about their* problem *and they get angry because they can't handle the problem.* **ID12, Jakarta, experience of mental illness.**

CYP with experience of mental illness were more likely to provide greater detail on the identified triggers of mental illness because they were able to draw on personal experience. They were also more likely to consider the interaction of multiple factors relating to the onset of mental illness and to acknowledge that causes and risk factors may vary for different people with different circumstances.

*I think there might be several reasons, or several things have happened to them in their lives, like war, or life problems like bad events, accidents—those can cause depression or anxiety. It might be hereditary or supernatural, but I'm not sure. Like when someone they loved died in an accident, they'll always miss them.* **ID10, Jakarta, experience of mental illness.**

*Q: What are the causes of mental health problems?*

*R*: *It depends on the person.* **ID41, Magelang, experience of mental illness.**

Identified stressors for mental illness identified by both groups included biological causes, social causes and traumatic experiences. On rarer occasions mental illness was referred to as a punishment from God or as a result of supernatural forces which compounded feelings of shame and embarrassment relating to the attribution of personal failing to mental illness.

*Supernatural things like ghosts can also make people depressed and anxious because they can make people feel afraid.* **ID30, Magelang, no experience of mental illness.**

*Q: What causes mental illness?*

*R: Trials from god*:

*Q: Trial? What kind of trial?*

*R: There's a lot of them, there's always a trial from God for us, it can come from ourselves, from our neighbours, family, surrounding, or our school friends.* **ID24, Bogor, experience of mental illness.**

Participants in both groups often attributed mental illness to deficits in brain chemistry. However, the two CYP groups diverged in their views on heritability of mental illness. CYP without a diagnosis of anxiety or depression expressed definitive opinions that this was not a risk factor for mental illness. CYP with a diagnosis of anxiety or depression were more likely to recognise the contributions of genetic factors particularly if they had a family member who also lived with a mental health problem.

*Q: Can it be hereditary?*

*R: There's no way it can be hereditary.* **ID19 Bogor, no experience of mental illness**

*Q: Why do mental health problems happen? For example, can they be hereditary?*

*R: Yes, from Dad. . . Dad also has mental health problems.* **ID13 Jakarta, experience of mental illness**

Social stressors in relation to mental illness were particularly salient for both CYP groups. These included family level influences such as strict and demanding parents, poor upbringing and absent or fractious relationships with family members or the wider community. Participants also acknowledged the role of friends in relation to the onset of mental illness citing the influence of negative role models but also highlighting the role of stress from relationships with friends that broke down or became challenging. Loneliness and social exclusion were also cited as important social causes of mental illness. This appeared particularly relevant to school situations which when combined with the stress of studying was considered a particularly salient risk factor for mental illness in the accounts from both CYP groups.

*It's [mental illness] also affected by their friends, so if they're mentally ill there's a high probability that they have a bad role model. Friends can be stressors: It's true that who you hang out with affect our intent and tenacity in life. Because having friends at school is really hard. It can be really different in just a month's time.* **ID28, Magelang, no experience of mental illness**

*Q: Why do you think that you've been experiencing sadness?*

*R: Because I've been extremely mocked.*

*Q: When you're experiencing that, what would make you feel worse?*

*R: If my friends keep going and don't stop.* **ID27, Bogor, experience of mental illness.**

Participants across both CYP groups identified similar social-interpersonal causes for mental illness which included the loss of loved ones through relationship breakdown or death and other traumatic experiences (e.g., immigration processes, sexual abuse, natural disaster).

*Q: Why do you think people become depressed or anxious? How does it happen?*

*R: Maybe something happened to them in the past and it's since happened again, like an earthquake. . . They're afraid that they'd go through another earthquake.* **ID17, Bogor, no experience of mental illness**

Bullying was the most commonly described environmental stressor with participants describing the negative effects of being mocked by others and the impact this could have on mental health.

*Q: Why do you think people are depressed or anxious? How does it happen?*

*R: Because other people tease them.* **ID40, Magelang, experience of mental illness**

*Q: Can you tell me how your mental health problems started?*

*R: I was teased, threatened.*

*Q: What did they say.*

*R: Well. . . "We'll wait for you at the school gate."* **ID41, Magelang, experience of mental illness.**

## Timeline dimensions

Timeline dimensions refer to whether people view their condition as chronic or acute. Participants in both groups rarely described potential or actual recovery in responses about mental illness and instead considered mental illness, particularly depression, as chronic, pervasive, never ending and strongly associated with negative consequences and hopelessness. This was in stark contrast to the time dimensions relating to beliefs about positive mental health and wellbeing described below.

*I'll say that it's a disease that's like a black hole. When we're going through it, it's hard to get out of it, and everything around us is dark. It feels like life goes on a loop every day, everything feels weary, nothing feels fresh anymore [. . .] Depression is a feeling that makes you feel sad. You don't know what to do with that feeling. Every day seems like the day before, everything seems dark, everything seems meaningless.* **ID10 Jakarta, experience of mental illness.**

*Depression is when someone feels that they're useless, like, ah, my life's just been like this forever, it's better to find another way. In the end they'd do something bad to themself.* **ID15, Bogor, no experience of mental illness.**

CYP described positive mental health as the outcome of connections to valued activities and creative pursuits and to supportive and functional familial and social relationships. In this way, the time dimensions of positive mental health were often acute. Concepts of positive mental health were often merged with those of positive affect and evidenced by peaks of emotion that correlated with discrete events rather than general periods of contentedness. The capacity to feel these peaks of emotion was taken as an essential indicator of mental health with little consideration given to what occurred between these events.

*Q: How did it feel when your dad took you to the fair ground?*

*R: Of course I was happy, I felt happy because my dad works in another town, so I don't meet him often.*

*Q: If you can feel a feeling of happiness, is that a sign that you're healthy?*

*R: Yes.* **ID27, Bogor, experience of mental illness.**

*Q: Why does this picture represent mental health to you?*

*R: It feels nice, it makes me feel happy. Enjoying nature with my family, so I feel more energized than when I'm home.* **ID26, Bogor, experience of mental illness.**

## Beliefs about curability

Personal responsibility and reluctance to seek help from others were highly salient in the accounts of CYP with and without anxiety or depression when discussing self-help interventions. Participants felt it was important to control their own emotions and improve individual character traits in order to manage their mental health. These dominant beliefs about personal responsibility could sometimes lead to CYPs downplaying or dismissing support from professional or lay support networks.

*Q: If you have a mental health problem, will you seek professional help? Why?*

*R: Probably I won't, we should be the one who puts up with our problem, don't let anyone that we love know about it, or they might become anxious too.* **ID1, Jakarta, no experience of mental illness.**

*Q: Do you think professionals—such as doctors, psychologists, psychiatrists—can help people with mental health problems to get better?*

*R: Errr. . . No, the only one who can help you is yourself.* **ID12, Jakarta, experience of mental illness.**

Distraction was identified by both groups of CYP as the primary way in which participants could manage their own mental health or provide help to others. Engaging in enjoyable activities, playing and spending time with friends and family particularly parents and siblings were most commonly implicated by participants in the study. This suggests an emerging belief that self-management skills can be developed.

We have to ask them to play. . . play tag. We do rock paper scissors and we tap them to be it

*As a group*, **ID4, Jakarta, no experience of mental illness.**

*R: Sometimes I like playing stuff by myself. Just playing things like, pretending I have an office, playing with dolls, pretending to cook.*

*Q: What's the benefit of playing them?*

*R: It makes me feel happy, relieved. It takes things off my mind.* **ID22, Bogor, experience of mental illness.**

CYP without experience of mental illness were more ambivalent towards professional help seeking compared to CYP with experience. There were two distinguishing features of responses from CYP without experience of mental illness. First, this participant group often described professional help seeking as acceptable only if the illness had reached a severity threshold. Secondly, CYP in this participant group were sceptical about treatment for anxiety or depression, and the efficacy of medication for the treatment of depression specifically. However, those with experience of anxiety or depression were slightly more likely to endorse the treatability of mental health problems and talking therapies in particular.

*Q: Do you think professionals are the best resource of help?*

*R: I don't think so. . . If it's too severe then yes. . .* **ID14, Bogor, no experience of mental illness**

*Q: Do you think professionals—such as doctors, psychologists, psychiatrists—can help people with mental health problems to get better? Especially depression and anxiety?*

*R: Yes, because they all studied for it, so they know how to handle it.* **ID10, Jakarta, experience of mental illness.**

## Beliefs about consequences

Beliefs about the consequences of mental illness amongst both groups had strong functional dimensions. Current perceptions of mental illness emphasized unusual visual or behavioural characteristics and were directly aligned to negative consequences relating to individual emotions, academic ability and performance or relationships with friends and family which could lead to social marginalisation and isolation. There were also associated burdens to family

members and communities reported by CYP in both groups. No CYP in either group reported any positive outcomes of the experience of mental illness. Instead, participants described the significant negative consequences of mental illness including hopelessness, homelessness, self-harm, suicide and social ostracization.

> *Yes. . . It becomes hard for them to communicate with others, and they get embarrassed in class, they'd spend time alone by themself.* **ID17, Bogor, no experience of mental illness.**

> *I've seen a crazy person by the highway. They were carrying some snacks, their clothes are torn, and their body was pitch black, and they didn't shower, their hair was smelly, and they themself were smelly.* **ID42, Magelang, experience of mental illness**

## Discussion

We conducted a qualitative analysis incorporating photo elicitation methodology underpinned by the Common Sense Model of Self-Regulation of Health and Illness [20] in order to develop an in-depth understanding of the mental health and illness perceptions of CYP aged 11–15 in Java, Indonesia. Drawing on an existing theoretical framework [20], our qualitative analysis helped elucidate the key features which characterise how CYP in Indonesia conceptualise positive mental health and illness and identify the illness perceptions which might most usefully be targeted with the greatest potential for impact. Our data also highlighted important and nuanced similarities and differences in these perceptions in relation to existing literature and the influence of lived experience (See S1 Fig) which have important implications for the targeting of future interventions.

Mental health has been defined by the World Health Organisation as "a state of well-being in which the individual realizes his or her own abilities, can cope with the normal stresses of life, can work productively and fruitfully, and is able to make a contribution to his or her community" [41]. This represents an important evolution from previous conceptualisations of mental health and illness as dichotomous entities although it does raise concerns about the applicability of such definitions for children and young people, whose primary focus may not be on occupational functioning and productivity [42]. Contrary to the WHO definition and recent models of mental health and illness which conceptualise two distinct continua [43], the CYP in the current study retained a dichotomous approach to mental health and illness, bluntly characterising the latter as the absence of the former. At the same time however, they foregrounded positive emotions and functionality as key features of mental wellbeing, more closely aligning their concepts of positive health to the current WHO definition [41].

As a result of adopting a functional emphasis within a dichotomous illness model, many CYP tended towards value-based judgments and the attribution of negative illness identities and stigmatising attitudes to those experiencing mental health difficulties. Such negative attributions have been demonstrated elsewhere in the literature [43–45]. None of the participants in our study reported any positive outcomes of mental illness, tending instead to characterise mental ill health as periods of absence of occupational and social functionality and creativity. These perceptions contrast with recent literature which demonstrates an association between mental health diagnoses and creativity [46] and reports of personal growth following mental illness which feature centrally in Western recovery models [47, 48]. Our findings lend support to previously highlighted limitations of definitions of health which are considered to place too much value on idealised norms and add to a growing call for more inclusive and dynamic definitions of mental health and illness which are better able to encapsulate the breadth of emotions that a person can experience during their life course [42]. Our work also highlights the

potential for the use of continua models of mental health and illness to target these illness beliefs including the development of mental illness concepts that allow people to move from illness to health and how this might effectively be done. Such approaches have been used effectively for adult populations to reduce stigma [49] but are likely to require cultural and developmentally appropriate adaptation prior to implementation.

Despite purposively sampling children and young people with a range of experiences, mental illness appeared at times to be of limited salience to participants [50]. This represents a key difference from existing literature undertaken with older adolescents in Indonesia [17]. This may because participants in the current study were younger and the identified lack of understanding reflected their earlier stage of development. CYP in the current study, particularly those with no experience of mental illness, did not appear to feel they initially had a clear understanding of mental illness or the requisite language required to articulate their thoughts on the topic. Participants found it easier to describe positive mental health and wellbeing, particularly when using the photographs they brought along to the interviews which provided access to the necessary components of their own vocabulary to enable this articulation. Prior research advocates the use of broader conceptualisations of mental health with CYP [29] and underscores of the potential value of using creative methods to optimise CYP engagement and develop effective interventions [51] which may be particularly useful in other mental health settings with less established health services.

Illness perceptions related to the identity, consequences and curability components of mental illness were dominated by negative perceptions of mental health problems. In high income countries, such dominance has been associated with higher ratings of unmet needs, poorer attitudes toward medication and adverse health outcomes [24], suggesting a potential intervention point for individual and community learning. Those with experience of mental illness were more likely to endorse the treatability of mental health conditions which points to the utility of developing community or peer delivered interventions which combine evidence with experiential and tacit based learning.

Interestingly CYP in both groups reported causal perceptions of mental illness as an interplay of external forces and individual character strengths, placing high emphasis on individual control in relation to the development and treatment of mental health problems. This may be a feature of the individual and moralistic character-building focus of the current Indonesian education system [52] as well as some aspects of wider Javanese culture, particularly 'rukun' (social harmony and communality) which discourages the upsetting or burdening of others [53]. This was reflected in CYP's apathetic outlook in terms of the trajectory of mental illness and the effectiveness of professional treatment and the foregrounding of distraction and engagement in valued activities as forms of self-help. This suggests an emerging belief amongst CYP that self-management strategies can be developed and implicates the role of voluntary sector in delivering self-care and civic interventions. These findings also demonstrate a need for and the potential value of implementing mental health literacy interventions in Indonesian community and educational contexts. Such interventions should focus on targeting identified illness beliefs such as individual weakness as a cause of mental illness, the view that mental illnesses are chronic and pervasive and the ambivalence towards professional treatment to optimise engagement with and the impact of the intervention.

The CSM is underpinned by the dynamic nature of illness perceptions which are considered responsive to changing circumstances and personal experiences [20] Whilst limited in nature, the study identified some important and nuanced differences in the views of those with and without experience of mental illness within CSM components–S1 Fig [43]. Mental illness was consistently conceived as a single homogenous entity manifesting in pervasively negative, abnormal and all-consuming emotions which led to social withdrawal and isolation. There

were few references to end points in descriptions of mental illness, although there was a suggestion that the direct, lived experience of mental illness had the potential to shape perceptions in the CSM's cause and perceived curability domains. These findings point to the potential cultural and development transferability of the CSM and lend support to previous research which demonstrates its applicability to CYP populations [30, 31]. However, future research could build on this exploratory work and test such hypotheses in more detail in prospectively designed longitudinal studies. Perceptions of protracted or more prolonged health threats have previously been associated with poorer functional outcomes [54], adding to the impetus to develop meaningful educational and intervention programmes for this group.

## Strengths and limitations

The study gains its strength from the combination of in-depth interviews, photo elicitation methodology and an existing theoretical framework. The photo elicitation method was a useful tool when combined with the broader conceptualisation of health and illness as it promoted more active participation in the interviews whilst ensuring that CYP views, thoughts and experiences guided interviews. Participants were able to raise issues and topics that were relevant and meaningful to them which may not have arisen in more structured interviews or through quantitatively designed studies.

The data presented in the study represents the views of children and young people aged 11–15 in Java Indonesia from a broad range of geographical areas and service settings (schools and formal health services) which have not been previously represented in existing literature. The views of other stakeholders (parents, health professionals, teachers and other key informants) have been collected using other methods and will be presented elsewhere. All data were collected by local researchers following detailed data collection protocols under the supervision of UK and Indonesian senior academics.

Our use of innovative methodologies combined with analytical approaches informed by an existing theoretical framework enabled the analysis to go further than available discourse in this area and explore the different dimensions of health and illness perspectives. The data presented here will inform the development and content of an intervention to support self-management and promote mental health literacy amongst CYP in Indonesia. This exploratory study points to the potential cultural and developmental transferability of the CSM to CYP populations and LMIC contexts which requires further examination.

All participants self-selected themselves for inclusion in the study and those with a diagnosis of mental illness had received some input from statutory or community services and therefore may have more positive views of formal service providers than those with no such contact. We also did not purposively sample on the basis of socio-economic status and did not collect demographic data in this regard. Future research should therefore further increase geographical representation across Indonesia from a range of socio-economic backgrounds and consider the inclusion of participants who have not had contact with formal services or have experienced different diagnoses such as Schizophrenia or Bipolar Disorder to incorporate wider perspectives on mental health and illness in Indonesia. Further work could be done to explore alternative ways to modify identified illness beliefs including direct and indirect interventions which draw on and remain cognizant of local community and parenting practices.

## Conclusion

The success of future mental health interventions in Indonesia and across LMICs is likely to rely heavily on the use of broad and inclusive definitions of mental health and illness and an in-depth understanding of individual health and illness perceptions to enable appropriate

targeting of interventions in relation to age, and past experience of mental illness. Public health education and targeted interventions to increase understanding may be necessary to develop alongside healthcare systems to reduce treatment barriers, optimise return on investment and enhance population health effect.

## Supporting information

**S1 Fig. Overview of the comparative analysis of CYP with and without experience of mental illness.**
(JPG)

**S1 File. Photo crib sheet.**
(PDF)

## Acknowledgments

The authors would like to thank the PPIE advisory panel for their feedback on emerging findings and support for the recruitment to the study. We would also like to acknowledge the participants for taking the time to take part in interviews without which this study would not have been possible. We would also like to thank KPSI, Into the Light and Pulih@the Peak for their supportive contributions to the project.

## Author Contributions

**Conceptualization:** Helen Brooks, Irmansyah, Benny Prawira, Karina Lovell, Ira Savitri Tanjun, Soraya Salim, Laoise Renwick, Penny Bee.

**Data curation:** Helen Brooks, Penny Bee.

**Formal analysis:** Helen Brooks, Kirsten Windfuhr, Irmansyah, Benny Prawira, Karina Lovell, Susi Rutmalem Bangun, Armaji Kamaludi Syarif, Christa Gumanti Manik, Laoise Renwick, Rebecca Pedley, Penny Bee.

**Funding acquisition:** Helen Brooks, Irmansyah, Benny Prawira, Karina Lovell, Ira Savitri Tanjun, Laoise Renwick, Penny Bee.

**Investigation:** Irmansyah, Benny Prawira, Dyah Afina Desyadi Putriningtyas, Susi Rutmalem Bangun, Armaji Kamaludi Syarif, Christa Gumanti Manik, Soraya Salim, Rebecca Pedley.

**Methodology:** Helen Brooks, Kirsten Windfuhr, Irmansyah, Dyah Afina Desyadi Putriningtyas, Ira Savitri Tanjun, Penny Bee.

**Project administration:** Helen Brooks, Irmansyah, Armaji Kamaludi Syarif, Christa Gumanti Manik, Ira Savitri Tanjun, Rebecca Pedley, Penny Bee.

**Resources:** Benny Prawira, Dyah Afina Desyadi Putriningtyas, Soraya Salim.

**Software:** Irmansyah.

**Supervision:** Helen Brooks, Kirsten Windfuhr, Irmansyah, Karina Lovell, Susi Rutmalem Bangun, Armaji Kamaludi Syarif, Christa Gumanti Manik, Ira Savitri Tanjun, Rebecca Pedley, Penny Bee.

**Validation:** Dyah Afina Desyadi Putriningtyas.

**Writing – original draft:** Helen Brooks, Kirsten Windfuhr, Irmansyah, Karina Lovell.

**Writing – review & editing:** Kirsten Windfuhr,  Irmansyah, Benny Prawira, Dyah Afina Desyadi Putriningtyas, Karina Lovell, Susi Rutmalem Bangun, Armaji Kamaludi Syarif, Christa Gumanti Manik, Ira Savitri Tanjun, Soraya Salim, Laoise Renwick, Rebecca Pedley, Penny Bee.

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
