## [Decision Letter · Decision Letter 0]

2 Nov 2021

PONE-D-21-04834Children and young people's beliefs about mental health and illness in Indonesia: A qualitative study informed by the Common Sense Model of Self-RegulationPLOS ONE

Dear Dr. Brooks,

Thank you for submitting your manuscript to PLOS ONE. After careful consideration, we feel that it has merit but does not fully meet PLOS ONE’s publication criteria as it currently stands. Therefore, we invite you to submit a revised version of the manuscript that addresses the points raised during the review process.

We look forward to receiving your revised manuscript.

Kind regards,

Bradford Dubik

Academic Editor

PLOS ONE

Journal Requirements:

2. Please note that in order to use the direct billing option the corresponding author must be affiliated with the chosen institute. Please either amend your manuscript to change the affiliation or corresponding author, or email us at plosone@plos.org with a request to remove this option.

3. Please provide additional details regarding participant consent. In the Methods section, please ensure that you have specified (1) whether consent was informed and (2) what type you obtained (for instance, written or verbal). If your study included minors, state whether you obtained consent from parents or guardians. If the need for consent was waived by the ethics committee, please include this information.

4. We note that Supporting information file 1 in your submission contain copyrighted images. All PLOS content is published under the Creative Commons Attribution License (CC BY 4.0), which means that the manuscript, images, and Supporting Information files will be freely available online, and any third party is permitted to access, download, copy, distribute, and use these materials in any way, even commercially, with proper attribution. For more information, see our copyright guidelines: http://journals.plos.org/plosone/s/licenses-and-copyright.

a. You may seek permission from the original copyright holder of the Figures in Supporting information file 1 to publish the content specifically under the CC BY 4.0 license. 

Reviewers' comments:

Reviewer's Responses to Questions

**Comments to the Author**

1. Is the manuscript technically sound, and do the data support the conclusions?

Reviewer #1: Yes

Reviewer #2: Partly

Reviewer #3: Yes

2. Has the statistical analysis been performed appropriately and rigorously? 

Reviewer #1: N/A

Reviewer #2: N/A

Reviewer #3: N/A

3. Have the authors made all data underlying the findings in their manuscript fully available?

Reviewer #1: Yes

Reviewer #2: Yes

Reviewer #3: No

4. Is the manuscript presented in an intelligible fashion and written in standard English?

Reviewer #1: Yes

Reviewer #2: Yes

Reviewer #3: Yes

5. Review Comments to the Author

Reviewer #1: - The introduction is somewhat lengthy. It is better to be a little brief.

- The methods section is well written and detailed.

- Although the presented results are correct based on the model, but in general, it has limited the expression of the results. The tendency to provide comparisons between the two groups in expressing the results has reduced the general opinions of individuals and has limited the comparison of the two groups.

- Thematic analysis could, in addition to this analysis, show the general aspects of the participants' opinions.

- In expressing the results, there is no reference to the results of analysis by photo elicitation method.

- The discussion section is well written and detailed.

Reviewer #2: Introduction

The introduction does a good job laying the groundwork for the study, as far as the extent of the problem and the need to compliment the focus on structural barriers with individual-level considerations, such as illness perception.

This section, however, is missing the research by Munson and colleagues, who over a decade ago were the first team to apply the CSM to adolescents/young adults, and those with mood disorders. Authors need to acknowledge this work; please see Munson, Narendorf, Ben-David & Cole, 2019; Munson, Floersch, & Townsend, 2009. These studies provide context for this study as they are also exploring illness and treatment understanding/attitudes among adolescents.

Method

How did you code “experience of mental illness”? Was this based on child self-report? Or parent report? Or a screening measure? Or a medical record?

Were the interview questions based on dimensions of the CSM, meaning was the study deductive, or was it inductive? Can the authors provide a sample question from the interview protocols to the reviewers/readers?

Sound, careful rigorous methods. No issues with the methods other than those indicated above.

Results

Emotional representations of illness, i.e., shame, is a dimension of the CSM. Why didn’t the authors consider some of the emotion data in relation to this construct?

In the section on “cause of illness” you suggest that those with a mental illness were more likely to consider the “interaction of multiple factors”. This is an important finding, especially in contrast to the idea that individuals are inherently weak, which you suggest participants reported. Authors need to support this result of multi-factorial causality with an empirical data element. Generally, this paper needs more data included in the manuscript. This is an area where the authors should also compare these results to other studies, i.e., See Munson et al., 2019 for comparison “cause” data among older adolescents/young adults.

Timeline dimension in CSM refers to whether individuals perceive their condition as chronic or acute. Please rephrase your results within this context.

It seems there are a few interesting results that differentiate between those with and without a illness/condition. This seems rather important; yet, in this version of the manuscript it is not that emphasized.

Discussion/Implications

The study has great value, with a very important point, from this reviewer’s point of view, being the high levels of stigma and no sense of recovery in the Indonesian context. What can the authors more specifically suggest for next steps to improve the subjective perspectives of adolescents? Can they develop school-based education/literacy programs? What do these data tell you/inform you about what topics might especially be important to educate individuals on, i.e., cause of mental illness (not a sign of individual weakness).

Overall, this study has potential. It needs to include more of the published research in the area in more detail. It also needs to add more empirical data elements (actual quotations) to support some of the statements in the results section.

Reviewer #3: In summary, this is an interesting and well-written paper dealing with an important public mental health issue, particularly the focus on young people in LMIC. The rationale for the study is convincing and the methods used are appropriate. However, there are some limitations that should be considered prior to publication:

Major limitations

• There is a methodological inconsistency between the authors’ proposition to focus on narratives and the interviewer’s semi-structured questions. These are different forms of data collection: While narrative approaches refer to biographical experiences and thus, (latent) meanings, semi-structured questions stimulate subjective attitudes or explicit norms. Thus, please check the methodological approach and address the approach and/or the instrument for data collection in more details. Please provide a table including interview prompts and/or questions.

• Closely related to this, some questions appear as rather closed and/or even suggestive, e.g. “is that a sign that you’re healthy? --- Yes” (line 419-20).

• Some assumptions are presented without sufficient supporting arguments or quotes, e.g. “This was in stark contrast to the time dimensions relating to beliefs about positive mental health and wellbeing” (line 425). Please alaborate on such assumptions and give quotes.

• The number of participants is relatively high for an in-depths qualitative study. However, since the findings are presented in the form of examples, it is difficult for readers to assess the dominance (or marginality) of CYPs’ arguments and opinions. Please provide a table including an overview on the themes and the (rough) numbers of relating quotes, e.g. category structure.

• What role does gender play in CYPs’ views on mental health and illness?

The discussion section appears as relatively unstructured and includes some (new) results.

o Children and young people's beliefs about mental health and illness in Indonesia: A

qualitative study informed by the Common Sense Model of Self-Regulation

Summarily, this is an interesting and well-written paper dealing with an important public mental health issue, particularly the focus on young people in LMIC. The rationale for the study is convincing and the methods are appropriate. However, there are some limitations that should be considered prior to publication:

Major limitations

• There is a methodological inconsistency between the authors’ proposition to ask for narratives and the interviewer’s semi-structured questions. These are different forms of data collection: While narratives refer to biographical experiences and thus, (latent) meanings, semi-structured questions stimulate subjective attitudes or explicit norms. It might be a difference between CYP explicit opinions and (latent) meanings. Thus, please check the methodological approach and specify the instrument for data collection in more details. Please provide a table including interview prompts and/or questions.

• Closely related to this, some questions appear as rather closed and/or even suggestive, e.g. “is that a sign that you’re healthy? --- Yes” (line 419-20)

• Some assumptions are presented without sufficient supporting arguments or quotes, e.g. “This was in stark contrast to the time dimensions relating to beliefs about positive mental health and wellbeing” (line 425). Please explain such assumptions and provide quotes.

• The number of participants is relatively high for an in-depths qualitative study. However, since the findings are presented in the form of examples, it is difficult for readers to assess the dominance (or marginality) of CYPs’ arguments and opinions. Please provide an overview on the themes and the (rough) number of relating quotes.

• What role does gender play in CYPs’ views on mental health and illness?

The discussion section is not structured clearly and includes some (new) results. Please

o Please provide a clearer picture of what was found in the study compared to what was found in other studies.

o What kind of differences and similarities were found in comparison with other study settings?

o Please provide more insight into the context of the results in terms of cultural aspects. What is rukun?

• I can hardly follow the author’s assumptions about CYP’ coherence (line 544-). Is this a finding? If so, please provide the details in the results section. However, I’m not sure if the method is appropriate to measure coherence.

Minor limitations:

• The authors describe that almost 50% of high school student experience depressive symptoms and 10% have a diagnosis. It would be helpful to provide findings from international studies in order to relate these numbers.

• Different age groups: While in the introductory section, the authors refer to the age group of 15-24, the study group includes CYP between 11 and 15. Please check this and give reasons for the selection of this age group.

• Since socio-economic status is known as a significant factor for developing mental health problems, it is surprising that these information is lacking. E.g. parental education or professional status. At least, this limitation should be mentioned and explained in the discussion.

• What is the (professional, training) background of the researchers/interviewers?

• Please describe the transcription and the translation processes in more detail. What are problems with regard to language translation? How were these problems addressed?

• What is meant by “marginalised in interviews” (line 303)?

• Explanations of what was originally planned (focus groups) is not really important to know for readers (599-602).

Please report your qualitative study in line with standards for reporting qualitative research.

6. PLOS authors have the option to publish the peer review history of their article (what does this mean?). If published, this will include your full peer review and any attached files.

Reviewer #1: No

Reviewer #2: No

Reviewer #3: No

---

## [Author Response · Author response to Decision Letter 0]

5 Nov 2021

Please see attached response to reviewers.

---

## [Decision Letter · Decision Letter 1]

18 Jan 2022

Children and young people's beliefs about mental health and illness in Indonesia: A qualitative study informed by the Common Sense Model of Self-Regulation

PONE-D-21-04834R1

Dear Dr. Brooks,

We’re pleased to inform you that your manuscript has been judged scientifically suitable for publication and will be formally accepted for publication once it meets all outstanding technical requirements.

Kind regards,

Tanya Doherty, PhD

Academic Editor

PLOS ONE

Additional Editor Comments (optional):

Reviewers' comments:

Reviewer's Responses to Questions

**Comments to the Author**

1. If the authors have adequately addressed your comments raised in a previous round of review and you feel that this manuscript is now acceptable for publication, you may indicate that here to bypass the “Comments to the Author” section, enter your conflict of interest statement in the “Confidential to Editor” section, and submit your "Accept" recommendation.

Reviewer #2: All comments have been addressed

2. Is the manuscript technically sound, and do the data support the conclusions?

Reviewer #2: Yes

3. Has the statistical analysis been performed appropriately and rigorously? 

Reviewer #2: N/A

4. Have the authors made all data underlying the findings in their manuscript fully available?

Reviewer #2: Yes

5. Is the manuscript presented in an intelligible fashion and written in standard English?

Reviewer #2: Yes

6. Review Comments to the Author

Reviewer #2: All of my concerns were addressed in the revision. The manuscript is well developed, technically sound and offers a solid contribution to the literature.

7. PLOS authors have the option to publish the peer review history of their article (what does this mean?). If published, this will include your full peer review and any attached files.

Reviewer #2: No

---

## [Editor Report · Acceptance letter]

24 Jan 2022

PONE-D-21-04834R1 

Children and young people's beliefs about mental health and illness in Indonesia: A qualitative study informed by the Common Sense Model of Self-Regulation 

Dear Dr. Brooks:

I'm pleased to inform you that your manuscript has been deemed suitable for publication in PLOS ONE. Congratulations! Your manuscript is now with our production department. 

Kind regards, 

on behalf of

Professor Tanya Doherty 

Academic Editor

PLOS ONE